# COVID-19 vaccine access and attitudes among people experiencing homelessness from pilot mobile phone survey in Los Angeles, CA

**Randall Kuhn**[1]*, **Benjamin Henwood**[2], **Alexander Lawton**[1], **Mary Kleva**[3], **Karthik Murali**[3], **Coley King**[4], **Lillian Gelberg**[5]

1 Department of Community Health Sciences, Jonathan and Karin Fielding School of Public Health, University of California Los Angeles, Los Angeles, California, United States of America, 2 Suzanne Dworak-Peck School of Social Work, University of Southern California, Los Angeles, California, United States of America, 3 Akido Labs, Los Angeles, California, United States of America, 4 Venice Family Clinic, Los Angeles, California, United States of America, 5 Department of Family Medicine, University of California Los Angeles, Los Angeles, California, United States of America

* kuhn@ucla.edu

**Data Availability Statement:** The underlying data files are available through the openICPSR data base at: https://doi.org/10.3886/E140701V3.

## Abstract

People experiencing homelessness (PEH) are at high risk for COVID-19 complications and fatality, and have been prioritized for vaccination in many areas. Yet little is known about vaccine acceptance in this population. The objective of this study was to determine the level of vaccine hesitancy among PEH in Los Angeles, CA and to understand the covariates of hesitancy in relation to COVID-19 risk, threat perception, self-protection and information sources. A novel mobile survey platform was deployed to recruit PEH from a federally qualified health center (FQHC) in Los Angeles to participate in a monthly rapid response study of COVID-19 attitudes, behaviors, and risks. Of 90 PEH surveyed, 43 (48%) expressed some level of vaccine hesitancy based either on actual vaccine offers (17/90 = 19%) or a hypothetical offer (73/90 = 81%). In bivariate analysis, those with high COVID-19 threat perception were less likely to be vaccine hesitant (OR = 0.34, $P$ = 0.03), while those who frequently practiced COVID-19 protective behaviors were more likely to be vaccine hesitant (OR = 2.21, $P$ = 0.08). In a multivariate model, those with high threat perception (AOR = 0.25, $P$ = 0.02) were less likely to be hesitant, while those engaging in COVID-19 protective behaviors were more hesitant (AOR = 3.63, $P$ = 0.02). Those who trusted official sources were less hesitant (AOR = 0.37, $P$ = 0.08) while those who trusted friends and family for COVID-19 information (AOR = 2.70, $P$ = 0.07) were more likely to be hesitant. Findings suggest that targeted educational and social influence interventions are needed to address high levels of vaccine hesitancy among PEH.

## Introduction

People experiencing homelessness (PEH), who have high rates of comorbid conditions more typical of individuals 15–20 years older than their chronological age [1–3], are extremely

**Funding:** Funding for this study was provided by University of California Office of the President Emergency COVID-19 Research Seed Funding Grant #R00RG2888, awarded to RK. Additional funding was provided by the Conrad N. Hilton Foundation by way of the USC Homelessness Policy Research Institute, Hilton Foundation Grant #18036, awarded to BH and RK. Authors MK and KM also received support from Akido Labs in the form of salaries. The funders had no additional role in the study design, data collection and analysis, decision to publish, or preparation of the manuscript. The specific role of each author is articulated in the 'author contributions' section.

**Competing interests:** Authors MK and KM are employees of Akido Labs. This does not alter our adherence to PLOS ONE policies on sharing data and materials. There are no patents, products in development or marketed products associated with this research to declare.

**Abbreviations:** AOR, adjusted odds ratio; COVID-19, coronavirus disease 2019; FQHC, Federally Qualified Health Center; OR, odds ratio; PEH, people experiencing homelessness; PHQ-4, patient health questionnaire-4.

susceptible to COVID-19 (coronavirus disease 2019), with higher risk of hospitalization and death from infection [4, 5]. Highly effective vaccines against SARS-CoV-2, the virus that causes COVID-19, may thus hold an outsized benefit for PEH, particularly those living in congregate settings such as shelters or unsheltered encampments that had previously seen COVID-19 outbreaks [6, 7]. Yet concerns persist about vaccine hesitancy among this population. Studies have already documented that populations with elevated risks of current and lifetime homelessness are hesitant to accept vaccines more generally, most notably African American, low income and low schooling populations [8–10]. Behavioral models of vaccine hesitancy highlight the complex role of threat perception, activation and trust in vaccine decisions [11, 12]. All of these concerns may be at play given the physical and mental health issues and social isolation facing PEH [13]. Yet few studies have documented vaccine hesitancy for any condition among PEH [14], and we know of no study that has addressed COVID-19 vaccine hesitancy in this population.

Using a unique, rapidly-deployed online survey of homeless patients of a Federally Qualified Health Center (FQHC) in Los Angeles, we describe levels of vaccine uptake and hesitancy, and address covariates of hesitancy in terms of COVID-19 vulnerability, threat perception, protection and information sources, along with demographic covariates.

## Data and methods

This study was designed as a pilot for a larger platform to address the challenge of gathering ongoing, longitudinal data from PEH through monthly online surveys. A university-based research team worked closely with an FQHC partner with strong homeless outreach and an active electronic health record system with a messaging platform. The analysis met all requirements of the Strengthening the Reporting of Observational Studies in Epidemiology (STROBE) guidelines. Because the study was designed prior to the development of a vaccine, analyses were not pre-specified in any protocols.

Potential patient participants were identified as homeless by the FQHC based on self-report from a patient questionnaire and/or the presence of an ICD-10 diagnosis code for homelessness at any point in their patient history (N = 3,145). A total of 1,537/3,145 (48.9%) clicked on the pre-screening survey. Respondents were screened as survey-eligible if they were age 18+, living in LA County, and met the US Department of Housing and Urban Development definition of homeless: "People who are living in a place not meant for human habitation, in emergency shelter, in transitional housing, or are exiting an institution where they temporarily resided." Of the 190 individuals meeting these criteria, 136 completed the baseline demographic survey (71%). The Month 3 survey that incorporated vaccine questions was completed by 90 respondents, for a 66% retention rate (90/136). Age/sex/race composition of the study population was compared to the source population of patients in the EHR system.

Once enrolled in the study, surveys were delivered through a HIPAA-compliant, cloud-based data collection platform that was designed to suit the capabilities of the study population, with extensive consultation with a lived expertise advisory group and testing with unhoused clients. Informed consent was conducted via the survey questionnaire, requiring affirmative consent before proceeding with the survey and providing complete informed consent documentation at the start of each survey. A 5-minute baseline demographic and risk factor survey was conducted December 2020 through January 2021. Monthly surveys lasted 15 minutes on average and included questions on COVID-19 risk perception, protective behaviors and information sources along with physical and mental well-being. The third monthly survey conducted February 15–26, 2021 added questions on vaccine uptake and acceptability.

Participants received financial incentives of $5 for the baseline and for each monthly survey. The study protocols were approved by the 1st author's university IRB.

## Dependent variables

Vaccine uptake was measured with a two-part question that first asked whether a respondent had been offered a vaccine, followed by a hesitancy question based on actual or hypothetical behavior. For those who had been offered a vaccine, individuals who did not accept the vaccine were coded as vaccine hesitant. Among those who had not been offered the vaccine, respondents were asked if they would take the vaccine if they were offered it, with possible responses of "yes," "no" or "prefer not to answer." Those who responded "no" or "prefer not to answer" were coded as vaccine hesitant.

## Independent variables

The initial baseline survey included self-reports of age, sex/gender (male/female) and race/ethnicity (White non-Hispanic, any Hispanic/Latino, Black non-Hispanic, other). Vulnerability to severe COVID-19 complications was assessed at baseline using self-reports of the CDC's list of underlying medical conditions (CDC). Sheltered/unsheltered status was measured in the monthly survey based on where the respondent slept the previous night. COVID-19 threat perception was measured using a modified 4-item adaptation of the Fear of COVID-19 scale [15], with "high threat perception" classified as responding "agree/strongly agree" to at least 3/4 questions. COVID-19 self-protective behavior was measured using a four-item index of how frequently the respondent wore a mask, washed their hands, stayed 6 feet from others, and avoided touching their face. Anxiety/depression was measured using the Patient Health Questionnaire-4 (PHQ-4), with moderate-severe psychological distress classified using the documented scoring system [16].

## Statistical analysis

After describing the univariate distribution for all dependent and independent variables, we conducted bivariate analysis of vaccine hesitancy in terms of all independent variables using two-tailed chi-square tests of differences in proportions and two-tailed t-tests of differences in means. We then estimated a multivariate model including all factors shown to be significant in bivariate analysis. All statistical analyses were performed in Stata 16. Due to the relatively small sample size, we report significance at both the 5% and 10% levels.

## Results

The mean age of the sample was 48.7 and 59% of respondents were female (Table 1). The sample was predominantly White non-Hispanic (49%), followed by Hispanic/Latino (18%), other (18%), and Black/African American non-Hispanic (9%). Most respondents were unsheltered (44%). More than half (52%) of respondents were coded as having moderate/severe psychological distress according to the PHQ-4 screening. Thirty three percent of respondents perceived COVID-19 as a high threat, and 42% reported high COVID-19 protective behavior. More than half reported trust in some official source (62%) or mass media (56%), while 42% reporting trusting personal information sources such as friends, family or social media.

Fig 1 shows that, of the 90 respondents in the sample, 17 (19%) were offered the vaccine, 10 of whom accepted. Among the 73 not offered the vaccine, 37 (51%) said they would take it if offered, 23 said they would not (32%), and 13 declined to answer (17%). Given these results, 43 (48%) expressed vaccine hesitancy, as defined above. Among those who rejected an offer of the

**Table 1. Summary statistics by COVID-19 vaccine hesitancy.**

| | No hesitancy (n = 47) | | Hesitancy (n = 43) | | Total (n = 90) | | |
|---|---|---|---|---|---|---|---|
| | mean | (95% CI) | mean | (95% CI) | mean | (95% CI) | P value[a] |
| **Age** | 49.8 | (45.9–53.7) | 47.6 | (43.4–51.7) | 48.7 | (45.9–51.5) | 0.43 |
| **Sex (female)** | 0.60 | (0.45–0.73) | 0.58 | (0.43–0.72) | 0.59 | (0.48–0.69) | 0.89 |
| **Race** | | | | | | | 0.83 |
| White (non-Hispanic) | 0.47 | (0.33–0.61) | 0.51 | (0.36–0.66) | 0.49 | (0.39–0.59) | |
| Black/African American (non-Hispanic) | 0.11 | (0.04–0.23) | 0.07 | (0.02–0.20) | 0.09 | (0.04–0.17) | |
| Hispanic/Latino | 0.15 | (0.07–0.28) | 0.21 | (0.11–0.36) | 0.18 | (0.11–0.27) | |
| Other (non-Hispanic) | 0.19 | (0.10–0.33) | 0.16 | (0.08–0.31) | 0.18 | (0.11–0.27) | |
| Unreported | 0.09 | (0.03–0.21) | 0.05 | (0.01–0.17) | 0.07 | (0.03–0.14) | |
| **Housing status (last night)** | | | | | | | 0.57 |
| Unsheltered | 0.47 | (0.33–0.61) | 0.42 | (0.28–0.57) | 0.44 | (0.34–0.55) | |
| Sheltered | 0.32 | (0.20–0.47) | 0.30 | (0.18–0.46) | 0.31 | (0.22–0.42) | |
| Doubled up/hotel | 0.13 | (0.06–0.26) | 0.23 | (0.13–0.38) | 0.18 | (0.11–0.27) | |
| Other | 0.09 | (0.03–0.21) | 0.05 | (0.01–0.17) | 0.07 | (0.03–0.14) | |
| **PHQ-4 Score** | 6.62 | (5.67–7.57) | 4.95 | (3.86–6.04) | 5.84 | (5.11–6.57) | 0.02 |
| **PHQ-4 Moderate/Severe** | 0.62 | (0.47–0.75) | 0.41 | (0.27–0.57) | 0.52 | (0.42–0.63) | 0.06 |
| **COVID-19 Threat Index (out of 4)[b]** | 2.11 | (1.64–2.57) | 1.24 | (0.82–1.67)[b] | 1.70 | (1.38–2.03) | 0.008 |
| I fear COVID more than anything else | 0.51 | (0.37–0.65) | 0.32 | (0.19–0.48) | 0.42 | (0.32–0.53) | 0.07 |
| I feel anxious when hearing about COVID | 0.55 | (0.41–0.69) | 0.37 | (0.23–0.52) | 0.47 | (0.36–0.57) | 0.08 |
| I'm more likely to get COVID than most | 0.49 | (0.35–0.63) | 0.17 | (0.08–0.32) | 0.34 | (0.25–0.45) | 0.002 |
| I'm more likely to get very sick from COVID than most | 0.55 | (0.41–0.69) | 0.39 | (0.25–0.55) | 0.48 | (0.37–0.58) | 0.13 |
| **COVID-19 Threat—High ($\geq 3$)** | 0.45 | (0.31–0.59) | 0.20 | (0.10–0.35) | 0.33 | (0.24–0.44) | 0.01 |
| **COVID-19 Protective Behavior Index (out of 4)[c]** | 1.66 | (1.27–2.05) | 2.15 | (1.66–2.63) | 1.89 | (1.58–2.19) | 0.12 |
| Always wash hands after bathroom, before eating | 0.47 | (0.33–0.61) | 0.55 | (0.39–0.69) | 0.51 | (0.40–0.61) | 0.45 |
| Always stay 6 feet apart from people I didn't live with | 0.45 | (0.31–0.59) | 0.50 | (0.35–0.65) | 0.47 | (0.37–0.58) | 0.62 |
| Always wear a mask | 0.51 | (0.37–0.65) | 0.60 | (0.44–0.74) | 0.55 | (0.44–0.66) | 0.41 |
| Always try not to touch mouth, nose, eyes, face | 0.19 | (0.10–0.33) | 0.46 | (0.32–0.62) | 0.32 | (0.23–0.42) | 0.006 |
| **COVID-19 Protective Behavior—High ($\geq 3$)** | 0.34 | (0.22–0.49) | 0.51 | (0.36–0.66) | 0.42 | (0.32–0.53) | 0.10 |
| **COVID-19 info from official sources** | 0.71 | (0.56–0.83) | 0.51 | (0.36–0.66) | 0.62 | (0.51–0.71) | 0.06 |
| **COVID-19 info from media** | 0.64 | (0.49–0.77) | 0.46 | (0.32–0.62) | 0.56 | (0.45–0.66) | 0.09 |
| **COVID-19 info from personal sources** | 0.36 | (0.23–0.51) | 0.49 | (0.34–0.64) | 0.42 | (0.32–0.53) | 0.21 |

[a]Tests for significance by vaccine acceptance. Reported *P* values correspond to chi-square tests for categorical variables and 2-tail t-tests for continuous variables .

[b]Individual COVID-19 Threat Index items refer to those who responded "Agree" or "Strongly Agree" to each statement, with high threat perception coded as responding "Always" or "Almost Always" to at least 3/4 items.

vaccine or stated that they would not get the vaccine if offered (n = 30), the most common reasons cited for vaccine hesitancy or refusal were fear of side effects (37%), wanting to have more information (30%), and rejection of all vaccines (27%) (Fig 2).

Bivariate analysis (Table 1) revealed no significant differences in vaccine hesitancy across any key demographic variables, including age, sex, race, and last-night housing status. Hesitant respondents scored as having lower PHQ-4 scores (4.95 vs. 6.62, *P* = 0.02) and were less likely to have moderate/severe psychological distress (41% vs 62% *P* = 0.06). Respondents classified as vaccine-hesitant scored lower on the COVID-19 threat index (1.24, compared to 2.11, *P* = 0.008) and were less likely to report high threat perception based on 3 out of 4 perceived threats (20% vs. 45%, *P* = 0.01). Hesitant respondents were not significantly more likely to engage in ≥3 of 4 reported COVID-19 protective behaviors (51% vs. 34%, *P* = 0.10), but were

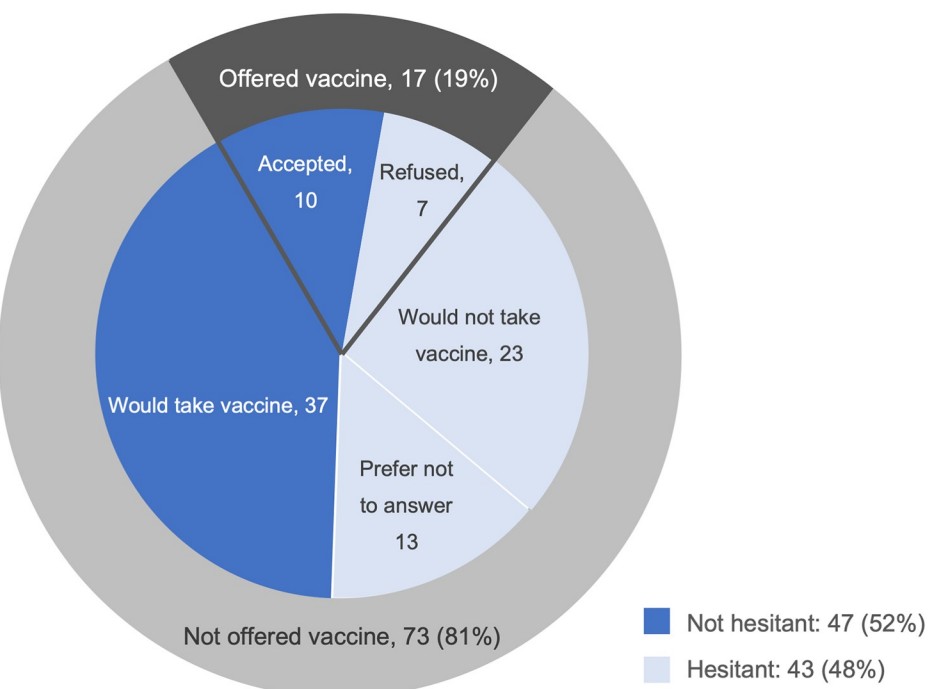

**Fig 1. COVID-19 vaccine hesitancy by prior vaccine access.** Respondents who were offered a vaccine (n = 17) were asked whether or not they received the vaccine; those who received the vaccine (n = 10) were classified as not vaccine hesitant and those who did not receive the vaccine were classified as vaccine hesitant (n = 7). Respondents who had not been offered the vaccine (n = 73) were asked if they would take the vaccine. Those who said they would take the vaccine (n = 37) were classified as not hesitant and those who said they wouldn't (n = 23) or declined to answer (n = 13) were classified as hesitant.

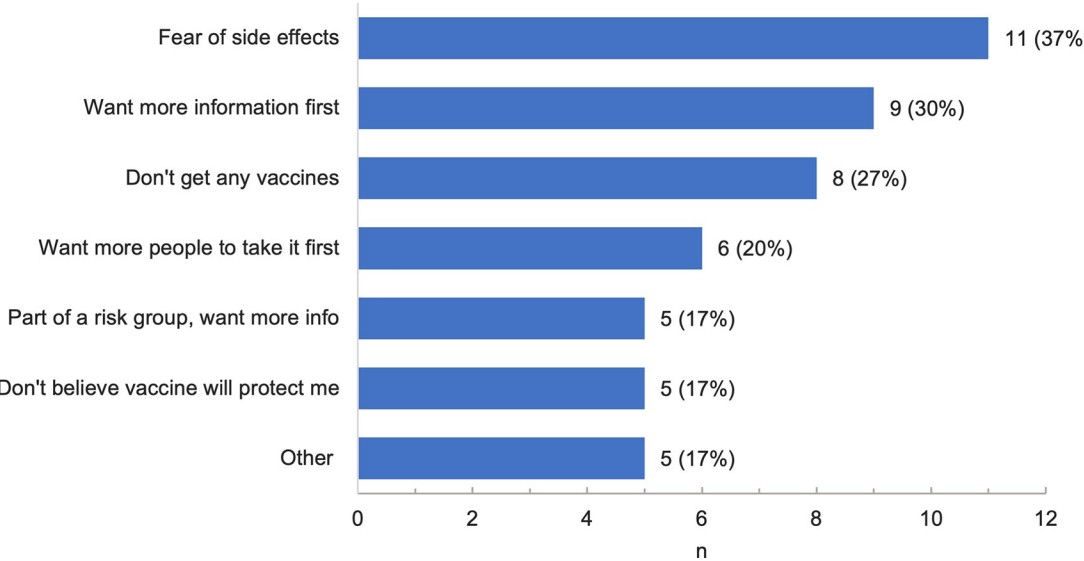

**Fig 2. Reasons for vaccine hesitancy among those who refused offer of COVID-19 vaccine.** Respondents who refused an actual or hypothetical offer of the COVID-19 vaccine were asked their reason(s) for refusal (n = 30). Other possible reasons for vaccine hesitancy that were not selected by any participants included "I am not at risk for COVID-19" and "I could not afford the vaccine".

**Table 2. Model of COVID-19 vaccine hesitancy.**

| Factor | OR | (95% CI) | P value | AOR | (95% CI) | P value |
|---|---|---|---|---|---|---|
| COVID-19 Threat Index—High | 0.34 | (0.13, 0.91) | 0.03 | 0.25 | (0.08, 0.80) | 0.02 |
| COVID-19 Protective Behavior—High | 2.21 | (0.92, 5.31) | 0.08 | 3.63 | (1.26, 10.47) | 0.02 |
| PHQ-4—Moderate/Severe | 0.49 | (0.21, 1.17) | 0.11 | 0.64 | (0.24, 1.71) | 0.38 |
| COVID-19 info from official sources | 0.41 | (0.17, 0.99) | 0.05 | 0.37 | (0.12, 1.11) | 0.08 |
| COVID-19 info from media | 0.50 | (0.21, 1.19) | 0.12 | 0.52 | (0.19, 1.41) | 0.20 |
| COVID-19 info from personal sources | 1.81 | (0.76, 4.32) | 0.18 | 2.70 | (0.93, 7.81) | 0.07 |
| n | | 85 | | | 85 | |
| Pseudo $R^2$ | | | | | 0.172 | |

significantly more likely to avoid touching their faces (46% vs. 19%, $P = 0.006$). They were less likely to trust COVID-19 protection information from official sources (51% vs. 71%, $P = 0.06$) or mass media (46% vs. 64%, $P = 0.09$) and no more likely to trust information from friends or social media (49% vs. 36%, $P = 0.21$).

A multivariate model showed that respondents with high COVID-19 threat perception were significantly less likely to be vaccine-hesitant (AOR = 0.25, $P = 0.02$) (Table 2). Those engaging in highly protective behavior were more likely to be vaccine-hesitant (AOR = 3.63, $P = 0.02$). Those trusting official sources were significantly less likely to be hesitant (AOR = 0.37, $P = 0.08$) and those trusting personal contacts more likely to be hesitant (AOR = 2.70, $P = 0.07$). A two-tailed t-test of equality in the coefficients for COVID-19 information sources revealed significantly higher levels of hesitancy for personal contacts vs. official sources (chi-square = 4.84, $P = 0.09$) and personal contacts vs. mass media (chi-square = 4.88, $P = 0.09$), with no significant difference between official sources vs. mass media.

## Discussion

Our findings provide initial evidence of high levels of hesitancy towards the COVID-19 vaccine among unhoused individuals. Based on a combination of actual and hypothetical behavior, 48% showed hesitancy toward the vaccine, considerably higher than the 31–35% observed in the general population over a similar period [9, 10]. The share who had been offered the vaccine was comparable to the general population of LA County at the time, and rates of hesitancy were nearly identical in actual and hypothetical responses.

Our findings point to the complex role of threat perception, activation and information in vaccine hesitancy among PEH. In adjusted models, respondents reporting higher COVID-19 fear were one quarter as likely to express vaccine hesitancy. At the same time, however, individuals who fully engaged in protective behaviors (e.g. mask-wearing) had nearly four times greater odds of vaccine hesitancy. Indeed, those who were hesitant towards the vaccine were more likely to engage in each of the four reported protective behaviors. This suggests that individuals who have actively engaged in COVID-19 protective measures over the past year may now be less accepting of the vaccine. Those who trusted COVID-19 information from official sources and news media were less hesitant, while those trusting personal sources (i.e. friends/ family and social media) were relatively more hesitant, which may also reflect a more general distrust of systems (e.g. health and homeless service systems) that are not designed to meet their needs. While we did not have sufficient power to test significance in reasons for hesitancy, we note that a higher proportion of those with high protective behavior reported reasons such as "I am part of a risk group and want more info," "I fear it will have unpleasant side effects," and "I do not believe the vaccine will protect me." Given that vaccine-hesitant

individuals are often more vocal in their beliefs, this points to the opportunity to leverage interpersonal networks as pathways of influence by focusing on individuals who may be especially activated or vocal about risks ascribed to both the disease itself and the vaccine [17].

This rapid-reaction pilot study has a number of limitations. First, the sample size was small and addressed patients only in one portion of West Los Angeles. Nevertheless, the results have been received as valuable to public health officials who are supporting additional enrollments across all service areas to increase the sample. Second, while all homeless-flagged patients with phones had the opportunity to answer the survey and response rates were considerably higher than most online or phone-based polls, we know that those who answered the survey were more likely to be female (59% vs. 35%) and less likely to be African-American (9% vs. 24%) than that clinic's homeless patient base as a whole. Given the lack of differences in hesitancy across any demographic groups and the small sample size, we did not report weighted results. Finally, we note that these interviews were conducted prior to PEH receiving universal vaccine eligibility on March 15, 2021, and that some hesitancy may more accurately reflect indifference or frustration at the difficulty of obtaining the vaccine.

In spite of these limitations, our findings point to challenges in widespread vaccination scaleup that are similar to those faced in the general population and in other marginalized populations. From a standpoint of individual risk, it is beneficial to know that the people who need the vaccine most—those who fear COVID-19 but are less likely to protect themselves through social distancing measures—are those most highly willing to be vaccinated. But in terms of achieving widespread vaccine acceptance and sub-population herd immunity, it may be far more challenging to achieve widespread uptake among individuals who are more proactive with protective behaviors, yet skeptical of the COVID-19 vaccine and less trusting of official information sources.

In order to bridge the gap between the portion of this highly vulnerable population that is nevertheless hesitant, it is likely that the health and social services sector will need to be able to demonstrate an increased ability to respond to the stated needs of this population. This will be critical to addressing the problem of homelessness while simultaneously promoting public health. While this pattern of widespread hesitancy may be unsurprising given the context, it nonetheless points to a potential pattern of syndemic risk observed in other highly vulnerable communities throughout the world [18]. In this particular case, the triple burden of chronic disease comorbidity, stigmatized exclusion from health systems, and pathogenic exposure create the potential conditions for recurrent epidemics and disease evolution [19]. The emergence of highly transmissible variants, combined with reduced attention to protective interventions, could lead to evolution in a subpopulation that has already been affected by recent outbreaks of Hepatitis A, tuberculosis, and typhus [20, 21]. Successful efforts to convince unhoused people of the benefits of vaccination, while requiring painstaking effort, could yield short-term benefits to both unhoused and housed communities, and promote resiliency to future pandemics and other disasters.

## Conclusion

Preliminary results from a small survey of PEH in Los Angeles reveal a high rate of vaccine hesitancy in this population, with higher levels of hesitancy observed among those with low threat perception, those engaging in self-protective behaviors, and those with higher trust in personal sources of information versus official sources. Our data suggest the need for targeted educational and social influence interventions to increase vaccine uptake among PEH, who are at greater risk of suffering from severe COVID-19 than the general population. Additional

data collected on a larger, more representative sample is necessary to determine differences in vaccine attitudes across demographic variables like race.

## Supporting information

**S1 File. STROBE statement—checklist of items that should be included in reports of** *cohort studies* **and enrollment diagram.**
(DOCX)

## Acknowledgments

The authors wish to thank the entire team at Venice Family Clinic for their partnerships, for reviewing the questionnaire, for distributing the survey to their patients, and for providing data support, specifically Karen C. Lamp, MD; Jenny O'Brien; Meghan Powers; Matthew Ware; and Carrie Kowalski. We would also like to thank the staff at Akido Labs who played an instrumental role in designing the survey and rapidly adding new questions, specifically Aishwarya Badanidiyoor. Finally, we are tremendously grateful to the support provided by our lived expertise group, a panel of individuals with past or current homelessness experience, for informing design of our study and research platform.

## Author Contributions

**Conceptualization:** Randall Kuhn, Benjamin Henwood, Karthik Murali, Coley King, Lillian Gelberg.

**Data curation:** Randall Kuhn, Alexander Lawton.

**Formal analysis:** Randall Kuhn, Alexander Lawton.

**Funding acquisition:** Randall Kuhn, Benjamin Henwood.

**Investigation:** Randall Kuhn, Benjamin Henwood, Karthik Murali.

**Methodology:** Randall Kuhn, Benjamin Henwood, Mary Kleva, Karthik Murali, Coley King.

**Project administration:** Randall Kuhn, Benjamin Henwood, Alexander Lawton, Mary Kleva.

**Supervision:** Karthik Murali, Lillian Gelberg.

**Writing – original draft:** Randall Kuhn, Alexander Lawton.

**Writing – review & editing:** Randall Kuhn, Benjamin Henwood, Alexander Lawton, Mary Kleva, Karthik Murali, Coley King, Lillian Gelberg.

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
