## [Decision Letter · Decision Letter 0]

11 May 2021

PONE-D-21-10720

COVID-19 vaccine access and attitudes among people experiencing homelessness from pilot mobile phone survey in Los Angeles, CA

PLOS ONE

Dear Dr. Kuhn,

Thank you for submitting your manuscript to PLOS ONE. After careful consideration, we feel that it has merit but does not fully meet PLOS ONE’s publication criteria as it currently stands. Therefore, we invite you to submit a revised version of the manuscript that addresses the points raised during the review process.

We look forward to receiving your revised manuscript.

Kind regards,

Prof. Anat Gesser-Edelsburg, Ph.D.

Academic Editor

PLOS ONE

Journal Requirements:

The authors have declared that no competing interests exist.

We note that one or more of the authors are employed by a commercial company: Akido Labs

3a.              Please provide an amended Funding Statement declaring this commercial affiliation, as well as a statement regarding the Role of Funders in your study. If the funding organization did not play a role in the study design, data collection and analysis, decision to publish, or preparation of the manuscript and only provided financial support in the form of authors' salaries and/or research materials, please review your statements relating to the author contributions, and ensure you have specifically and accurately indicated the role(s) that these authors had in your study. You can update author roles in the Author Contributions section of the online submission form.

3b. Please also provide an updated Competing Interests Statement declaring this commercial affiliation along with any other relevant declarations relating to employment, consultancy, patents, products in development, or marketed products, etc. 

Reviewers' comments:

Reviewer's Responses to Questions

**Comments to the Author**

1. Is the manuscript technically sound, and do the data support the conclusions?

Reviewer #1: Yes

Reviewer #2: Yes

2. Has the statistical analysis been performed appropriately and rigorously? 

Reviewer #1: Yes

Reviewer #2: Yes

3. Have the authors made all data underlying the findings in their manuscript fully available?

Reviewer #1: Yes

Reviewer #2: Yes

4. Is the manuscript presented in an intelligible fashion and written in standard English?

Reviewer #1: Yes

Reviewer #2: Yes

5. Review Comments to the Author

Reviewer #1: The article is very well written and of good quality.

Small points must be adjusted:

- include the clinical implications of the study, at the end of the discussion, after the limitations of the study.

- bring a little perspective of this reality in other countries, as is the vaccination situation of this population in other parts of the world.

Reviewer #2: This was a great article examining COVID-19 attitudes among people experiencing homelessness. The manuscript was well organized. The tables were excellent in providing both the statistical data but also in providing additional details about specific questions asked in the survey.

6. PLOS authors have the option to publish the peer review history of their article (what does this mean?). If published, this will include your full peer review and any attached files.

Reviewer #1: No

Reviewer #2: No

---

## [Author Response · Author response to Decision Letter 0]

15 Jun 2021

We have expanded the discussion to address the points raised by Reviewer #1, namely: we expanded upon the clinical implications of the study, specifically noting the historic impact of communicable diseases within homeless encampments and shelters, and the importance of vaccine acceptance beyond COVID-19 (see lines 247-255). We also addressed the global implications of our findings (see lines 245-247), in drawing parallels to people experiencing homeless with other vulnerable populations around the world.

We also addressed the following points raised by the academic editor:

• Ensured our manuscript meets PLOS ONE’s style requirements including those for file naming 

• Published our minimal data set underlying the results

• Revised our funding statement to address our affiliation with Akido Labs

• Revised our competing interests statement to address our affiliation with Akido Labs

• Expanded our reference list (added #18-21) to cite the points that were made in response to Reviewer #1’s requested changes

Included specific funding information for the grant from the Hilton Foundation (which was disbursed by the Homelessness Policy Research Institute)

Included a statement that we have no patents or products associated with this research to declare.

---

## [Decision Letter · Decision Letter 1]

13 Jul 2021

COVID-19 vaccine access and attitudes among people experiencing homelessness from pilot mobile phone survey in Los Angeles, CA

PONE-D-21-10720R1

Dear Dr. Kuhn,

We’re pleased to inform you that your manuscript has been judged scientifically suitable for publication and will be formally accepted for publication once it meets all outstanding technical requirements.

Kind regards,

Prof. Anat Gesser-Edelsburg, Ph.D.

Academic Editor

PLOS ONE

Additional Editor Comments (optional):

Reviewers' comments:

Reviewer's Responses to Questions

**Comments to the Author**

1. If the authors have adequately addressed your comments raised in a previous round of review and you feel that this manuscript is now acceptable for publication, you may indicate that here to bypass the “Comments to the Author” section, enter your conflict of interest statement in the “Confidential to Editor” section, and submit your "Accept" recommendation.

Reviewer #1: All comments have been addressed

Reviewer #2: All comments have been addressed

2. Is the manuscript technically sound, and do the data support the conclusions?

Reviewer #1: Yes

Reviewer #2: Yes

3. Has the statistical analysis been performed appropriately and rigorously? 

Reviewer #1: Yes

Reviewer #2: Yes

4. Have the authors made all data underlying the findings in their manuscript fully available?

Reviewer #1: Yes

Reviewer #2: Yes

5. Is the manuscript presented in an intelligible fashion and written in standard English?

Reviewer #1: Yes

Reviewer #2: Yes

6. Review Comments to the Author

Reviewer #1: The authors made all the adjustments that were requested. Congratulations on the article, it will make a big difference to society.

Reviewer #2: The authors addressed did a great job of addressing reviewer comments. Overall this manuscript provides important implications for those experiencing homelessness.

7. PLOS authors have the option to publish the peer review history of their article (what does this mean?). If published, this will include your full peer review and any attached files.

Reviewer #1: **Yes: **Mateus Dias Antunes

Reviewer #2: No

---

## [Editor Report · Acceptance letter]

22 Jul 2021

PONE-D-21-10720R1 

COVID-19 vaccine access and attitudes among people experiencing homelessness from pilot mobile phone survey in Los Angeles, CA 

Dear Dr. Kuhn:

I'm pleased to inform you that your manuscript has been deemed suitable for publication in PLOS ONE. Congratulations! Your manuscript is now with our production department. 

Kind regards, 

on behalf of

Prof. Anat Gesser-Edelsburg 

Academic Editor

PLOS ONE